# First Principle Investigation of Electronic, Transport, and Bulk Properties of Zinc-Blende Magnesium Sulfide

**Uttam Bhandari [1], Blaise Awola Ayirizia [2], Yuriy Malozovsky [2], Lashounda Franklin [2] and Diola Bagayoko [2,\***

[1] Department of Computer Science, Southern University and A&M College, Baton Rouge, LA 70813, USA; uttam.bhandari@sus.edu

[2] Department of Mathematics and Physics, Southern University and A&M College, Baton Rouge, LA 70813, USA; blaiseawola45@gmail.com (B.A.A.); yuriym802000@gmail.com (Y.M.); lfranklin2002@aol.com (L.F.)

\* Correspondence: bagayoko@aol.com; Tel.: +1-225-771-2730

**Abstract:** We have studied electronic, structural, and transport properties of zinc-blende magnesium sulfide (zb-MgS). We employed a local density approximation (LDA) potential and the linear combination of atomic orbitals (LCAO) method. Our computational method is able to reach the ground state of a material, as dictated by the second theorem of density functional theory (DFT). Consequently, our findings have the physical content of DFT and agree with available, corresponding experimental ones. The calculated band gap of zb-MgS, a direct gap equal to 4.43 eV, obtained at the experimental lattice constant of 5.620 Å, completely agrees with the experimental band gap of 4.45 ± 0.2 eV. We also report total (DOS) and partial (pDOS) densities of states, electron and hole effective masses, the equilibrium lattice constant, and the bulk modulus. The calculated pDOS also agree with the experiment for the description of the states at the top and the bottom of the valence and conduction bands, respectively.

**Keywords:** density functional theory; magnesium sulfide; local density approximation; linear combination of atomic orbitals; band gap; band structure; transport properties

## 1. Introduction

There are three (3) polymorphs of magnesium sulfide, i.e., rock salt, zinc-blende, and wurtzite structures. The zinc blende (zb) structure of MgS can be prepared by metal-organic vapor-phase epitaxy [1–3] or molecular beam epitaxy [4,5] by growing on a GaAs or ZnSe substrate. This material has been attracting much interest, in experimental and theoretical studies, due to its wide band gap and its technological applications. MgS has applications in optoelectronics, thin-film luminescent devices, blue light-emitting diodes, ultraviolet wavelength optics, optical erasable memory devices, and solar-blind UV detection devices [6–9]. The zb-MgS crystal has lattice matches with substrates like GaAs and ZnSe. It can also be used as an electrode in lithium-ion batteries and rechargeable magnesium sulfur batteries [10,11]. Furthermore, it is utilized as an excellent barrier material for quantum well structures [12].

Drief et al. [13] calculated the direct band gap of zb-MgS to be 3.371 eV, using the full potential linearized augmented plane wave (FP-LAPW) method. Various computational research groups [14–16] calculated the direct band gap of zb-MgS to be in a range of 3.10 eV to 3.46 eV with LDA potentials. Several other groups [17–20] have computed the electronic properties of zb-MgS with generalized gradient approximation (GGA) potentials. Their calculated, direct band gap value for zb-MgS was

in a range of 3.33 eV to 3.60 eV. Recently, Tairi et al. [18] performed calculations using the FP-LAPW computational method with the modified Becke-Johnson (mBJ) potential. They found a direct band gap for zb-MgS of 5.193 eV. The experiment performed by Okuyama et al. [21], i.e., X-ray diffraction (XRD) measurements, found a direct band gap of 4.45 ± 0.2 eV. Another measured value of the band gap of zb-MgS is 4.4 eV, as reported by Teraguchi et al. [4] who employed a low angle XRD method. The disagreement between computational works with ab-initio DFT potentials, and available experimental results partly motivated this study. The many current and potential applications of zb-MgS, as illustrated above, further add to our motivation for the present computations. We summarize below, in Table 1, findings from previous calculations with *ab-initio* and ad hoc DFT potentials.

**Table 1.** Calculated, direct band gap ($E_g$) of zb-MgS, using different computational techniques and potentials. The last two (2) rows show experimental band gaps. Pertinent reference numbers are in a superscript in Column III.

| Computational Technique | Potential | Direct Band Gap, $E_g$ (eV) |
| --- | --- | --- |
| FP-LAPW | LDA | 3.37 [13] |
| Plane wave pseudopotential approach | LDA | 3.10 [14] |
| Ab-initio | LDA | 3.42 [15] |
| Full Potential Linearized Augmented plane wave Method (FP-LMTO) | LDA | 3.46 [16] |
| Plane wave pseudopotential method | GGA | 3.38 [17] |
| FP-LAPW | GGA | 3.362 [18] |
| FP-LAPW | GGA | 3.33 [19] |
| FP-LMTO | GGA | 3.37 [16] |
| FP-LAPW | EV-GGA | 3.60 [20] |
| FP-LAPW | WC-GGA | 3.20 [20] |
| FP-LAPW | mBJ | 5.193 [18] |
| Full multiple scattering method, Muffin Tin (MT) | Crystal MT potential, with touching spheres | 4.6 ± 0.3 [21] |
| Modified dielectric theory | Not Applicable | 4.62 [22] |
| Photoluminescence measurements of zb-MgS thin film | - | 4.8 [5] |
| Experimental using XRD measurement | - | 4.45 ± 0.2 [22] |
| Experiment using low angle XRD measurement | - | 4.4 [4] |

Engel and Vosko generalized-gradient approximation (EV-GGA). Wu-Cohen generalized-gradient approximation (WC-GGA).

## 2. Computational Method

Our calculations employed the local density approximation (LDA) potential developed by Ceperley and Alder [23] and parameterized by Vosko et al. [24]. A detailed description of our calculation method is available in previous publications [25–34]. Known as the Bagayoko, Zhao, and Williams (BZW) [25–29] method, enhanced by Ekuma and Franklin (BZW-EF) [30,31,34], this method is characterized by the following features. (a) It leads to the true ground state of the material and (b) it does so while avoiding basic sets that are over-complete for the description of the ground state. We start our implementation of the linear combination of atomic orbitals (LCAO) formalism with a small basis set that accounts for all the electrons in the system/the difference between BZW and BZW-EF stems from the order [30,31] in which orbitals are added, one at a time, to augment the basis set.

The first self-consistent calculation is followed by a second whose basis set has one additional orbital. Depending on the s, p, or d nature of this orbital, the size of the new basis set will be larger than that of the initial one by 2, 6, or 10 functions, respectively, taking the spin into account. The comparison of the occupied energies of these first two calculations generally shows that the occupied energies from the second calculation (II) are lower than or equal to their corresponding values from Calculation I. The self-consistent Calculation III is performed using the basic set of Calculation II as augmented by one orbital. Again, we compare the occupied energies of Calculations II and III. This process continues until three (3) consecutive calculations lead to the same occupied energies. The perfect superposition of the occupied energies is the robust proof of the attainment of the ground state of the material. The first of these three calculations is the one that provides the DFT description [25,30–32] of the material. The corresponding basis set is the optimal basis set, i.e., the smallest basis set leading to



the ground state of the material. Hence, the optimal basis set, upon the attainment of self-consistency, produces the ground state charge density of the material [25]. Following the above description of our method, we provide details of our calculations.

The crystal structure of MgS is from the space group Fm$\bar{3}$m. The radial parts of the atomic orbitals comprise Gaussian functions. The atomic wave functions we employed for $Mg^{2+}$ had 18, 16, and 16 Gaussian functions for the s, p, and d orbitals, respectively. Likewise, the atomic wave functions for $S^{2-}$ had 18, 18, and 16 Gaussian functions for the s, p, and d orbitals, respectively. For the $Mg^{2+}$ ion, the Gaussian exponents, in atomic units, ranged from 0.1822 to $0.11 \times 10^6$ and those for the $S^{2-}$ ion ranged from 0.1489 to $0.44 \times 10^5$. A mesh of 81-k points was utilized in the Brillouin zone. The self-consistent calculations converged after 60 iterations. The criterion for convergence is a difference of $10^{-5}$ or less between the potentials from two consecutive iterations.

In this work, we used the LCAO program package [35,36] developed at the Ames Research Laboratory of the US Department of Energy (DOE), Ames, Iowa. While the package can be used to study chemical structures, we have mostly utilized it to study crystalline structures with the emphasis on semiconductors and insulators.

## 3. Results

### 3.1. Electronic Properties

The successive calculations performed in our generalized minimization of the energy are below in Table 2. Columns 2 and 3 display the specific valence orbitals on $Mg^{2+}$ and $S^{2-}$, respectively. The total number of valence functions and the resulting band gaps are, respectively, in columns 4 and 5.

**Table 2.** Successive calculations in the BZW-EF method, for zb-MgS, at a room temperature experimental lattice constant of 5.62 Å [4,22]. The highlighted Calculation IV led to the absolute minima of the occupied energies, i.e., the ground state. A superscript of zero denotes an unoccupied orbital.

| Calculation No. | Magnesium ($Mg^{2+}$) ($1s^2$-Core) | Sulfur ($S^{2-}$) ($1s^2 2s^2 2p^2$-Core) | No. of Valence Functions | Energy Gap (eV) |
|---|---|---|---|---|
| I | $2s^2 2p^6 3p^0$ | $3s^2 3p^6$ | $2 \times (7 + 4) = 22$ | 7.658 (Γ-X) |
| II | $2s^2 2p^6 3p^0$ | $3s^2 3p^6 4p^0$ | $2 \times (7 + 7) = 28$ | 7.001 (Γ-X) |
| III | $2s^2 2p^6 3p^0 3s^0$ | $3s^2 3p^6 4p^0$ | $2 \times (8 + 7) = 30$ | 6.414 (Γ-Γ) |
| IV | $2s^2 2p^6 3p^0 3s^0$ | $3s^2 3p^6 4p^0 4s^0$ | $2 \times (8 + 8) = 32$ | 4.435 (Γ-Γ) |
| V | $2s^2 2p^6 3p^0 3s^0 4p^0$ | $3s^2 3p^6 4p^0 4s^0$ | $2 \times (11 + 8) = 38$ | 4.435 (Γ-Γ) |
| VI | $2s^2 2p^6 3p^0 3s^0 4p^0 4s^0$ | $3s^2 3p^6 4p^0 4s^0$ | $2 \times (12 + 8) = 40$ | 4.419 (Γ-Γ) |

Figure 1 displays the band structure of zb-MgS as produced by Calculations IV (full lines) and V (dashed lines). The occupied energies are perfectly superimposed. Calculation IV produced the same occupied energies. As per the description of our method above, Calculation IV produced the DFT description of zb-MgS. Its basis set is the optimal one. The unoccupied energies from the two calculations are also superimposed up to 5 eV. For higher energies, as explained above, a given unoccupied eigenvalue from Calculation V is lower than or equal to the corresponding one from Calculation IV.

Even though the occupied energies of Calculations V and VI were the same as the corresponding ones from Calculation IV, some unoccupied energies from these calculations were lower than their corresponding values produced by Calculation IV, using the optimal basis set. This lowering of unoccupied energies, while the occupied ones do not change, is a mathematical artifact explained by the Rayleigh theorem [25]. The top of the valence band and the bottom of the conduction band of zb-MgS are at the Γ point. The calculated direct band gap of zb-MgS, using the optimal basis set of Calculation IV, is 4.43 eV. This band gap agrees with the available experimental band gap of $4.45 \pm 0.2$ eV [4,22]. Table 3 lists the calculated, eigenvalues in the range of −11.360 to +18.933 eV, at high symmetry points in the Brillouin zone. The content of this table lends itself to comparisons with future experimental findings, including those from Ultraviolet (UV) and X-ray spectroscopic studies.

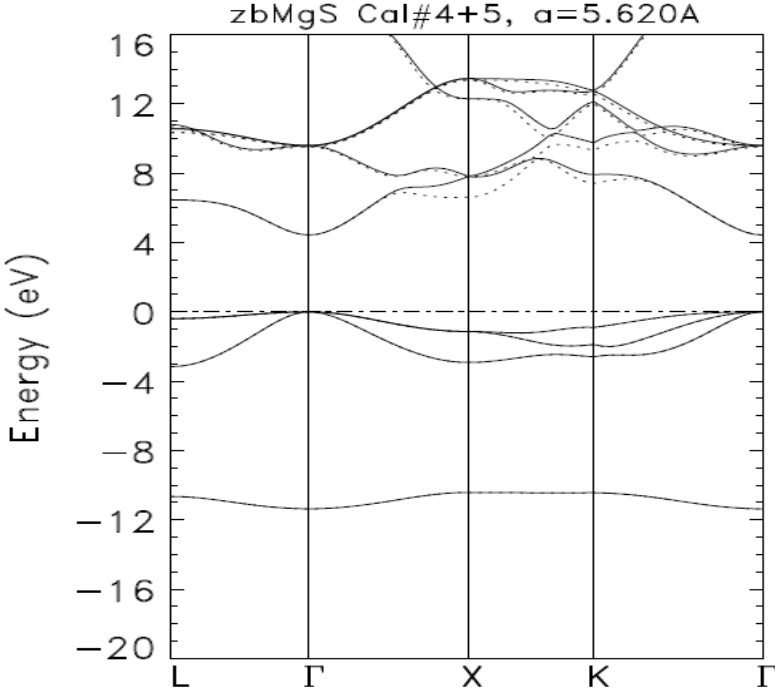

**Figure 1.** Electronic band structures of zb-MgS, as obtained from Calculations IV (—) and V ( ... ), using the BZW-EF method. The horizontal line, at zero, indicates the location of the Fermi level (EF).

**Table 3.** Calculated eigenvalues (in eV) for zb-MgS, as produced by Calculation IV of the BZW-EF method with a lattice constant of 5.6200 Å at room temperature.

| L-Point | Γ-Point | X-Point | K-Point |
|---|---|---|---|
| 18.933 | 16.064 | 13.458 | 12.770 |
| 10.806 | 9.595 | 13.458 | 12.736 |
| 10.566 | 9.595 | 12.291 | 12.128 |
| 10.566 | 9.595 | 7.831 | 9.755 |
| 6.464 | 4.435 | 7.791 | 7.919 |
| −0.390 | 0.000 | −1.135 | −0.899 |
| −0.390 | 0.000 | −1.135 | −1.884 |
| −3.157 | 0.000 | −2.917 | −2.592 |
| −10.648 | −11.360 | −10.415 | −10.428 |

The calculated, total (DOS) and partial (pDOS) densities of states of zb-MgS, in the energy range of −16 to 20 eV, are in Figure 2a,b, respectively. The highest peak in the DOS for the valence states is around the bottom of the valence band. The relative flatness of the lowest valence band explains this peak. The gap of 4.43 eV is between the top of the valence bands and the bottom of the conduction bands. The inset shows the magnified DOS within the energy range −1 eV to 7 eV. This 20-fold magnification shows the sharpness of the absorption edge between 4 and 5 eV.

In the pDOS, in Figure 2b, the lowermost portion of the valence bands is mostly from S-s with a tiny contribution from Mg-s. The uppermost portion of the valence band is largely contributed by S-p states with very small contributions from Mg-p and Mg-s states. The lowermost portion of the conduction band is mostly contributed by S-s and Mg-s states with minor contributions from S-p states.

Our pDOS results agree with some experimental ones from Kravtsova et al. [37]. These authors found that the S-p states lie at the top of the valence band of zb-MgS, as shown in our calculated pDOS above. Similarly, the calculated pDOS show that the bottom of the conduction band is S-s states and Mg-s states, as found experimentally [37].

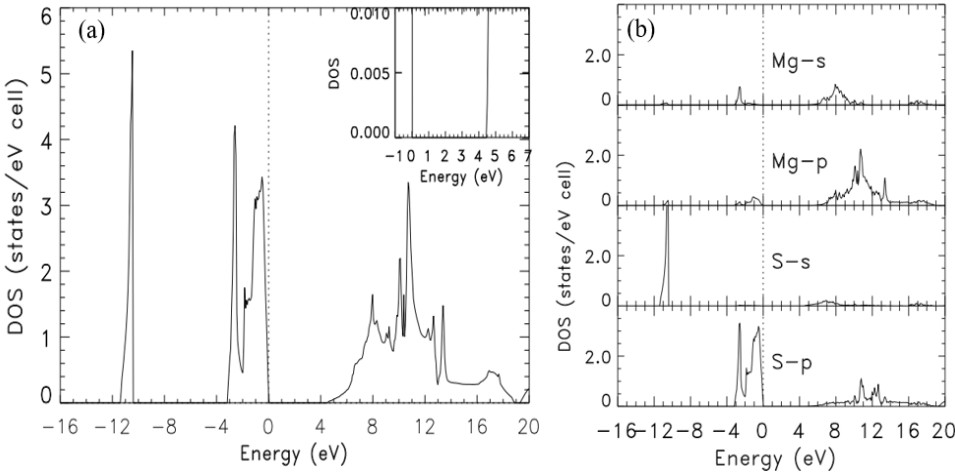

**Figure 2.** (**a**) The total density of states of zb-MgS, derived from the bands produced by Calculation IV. The insert shows a 20-fold magnification of the DOS around the band gap. (**b**) Partial densities of states (pDOS) of zb-MgS, derived from the bands produced by Calculation IV. The vertical, dashed line indicates the position of the Fermi level.

### 3.2. Transport Properties

Effective masses are needed for the calculation of several transport properties, including charge mobilities that are inversely proportional to these masses. Table 4 shows our calculated electron and hole effective masses for zb-MgS. Our calculations qualitatively confirm the findings of Deguchi et al. [38] that the electron effective mass is nearly isotropic, while, that for the hole, is anisotropic. This anisotropy is much more pronounced for the two heavy holes as compared to the light holes. Deguchi et al. [38] calculated the effective mass of zb-MgS using the quasiparticle self-consistent GW method. They reported average effective masses of 0.248 $m_o$, 1.261 $m_o$, and 0.248 $m_o$, for the electron ($M_e$), heavy hole ($M_{hh}$), and light hole ($M_{lh}$), respectively where $m_o$ stands for free electron mass. The calculation performed by the Kumano et al. [39], using a fitting approach, found average effective masses of $M_e = 0.27\,m_o$, $M_{hh} = 0.49\,m_o$, and $M_{lh} = 0.49\,m_o$. These previous, calculated effective masses are systematically lower than our corresponding results in Table 4. We propose an explanation of this difference in the Discussion section.

**Table 4.** Calculated, effective masses for zb-MgS, in units of free electron mass ($m_o$): $M_e$ denotes an electron effective mass. $M_{hh}$ and $M_{lh}$ represent the heavy and light hole effective masses, respectively.

| Types and Directions of Effective Masses | Values of Effective Masses (mo) |
|---|---|
| Me (Γ-L) 111 | 0.306 |
| Me (Γ-X) 100 | 0.317 |
| Me (Γ-K) 110 | 0.314 |
| Mhh1 (Γ-L) 111 | 3.255 |
| Mhh1 (Γ-X) 100 | 1.446 |
| Mhh1 (Γ-K) 110 | 2.035 |
| Mhh2 (Γ-L) 111 | 3.025 |
| Mhh2 (Γ-X) 100 | 1.446 |
| Mhh2 (Γ-K) 110 | 1.595 |
| Mlh (Γ-L) 111 | 0.309 |
| Mlh (Γ-X) 100 | 0.408 |
| Mlh (Γ-K) 110 | 0.371 |

### 3.3. Structural Properties

Figure 3 exhibits the computed, total energy as a function of the lattice constant. The minimum total energy lies at a lattice constant of 5.56 Å, which is our computed equilibrium lattice constant at zero Kelvin. Our calculation of the bulk modulus first entailed a least square fitting of the total energy curve in the vicinity of its minimum. We obtained the bulk modulus from the second derivative of

this fit at the equilibrium lattice constant, i.e., B = V d$^2$E/dV$^2$. We found a bulk modulus of 60 GPa. Other theoretical calculations [13,16,40–42] also obtained the bulk modulus in a range from 60 GPa to 63 GPa. We are not aware of a reported, experimental value for the bulk modulus of zb-MgS.

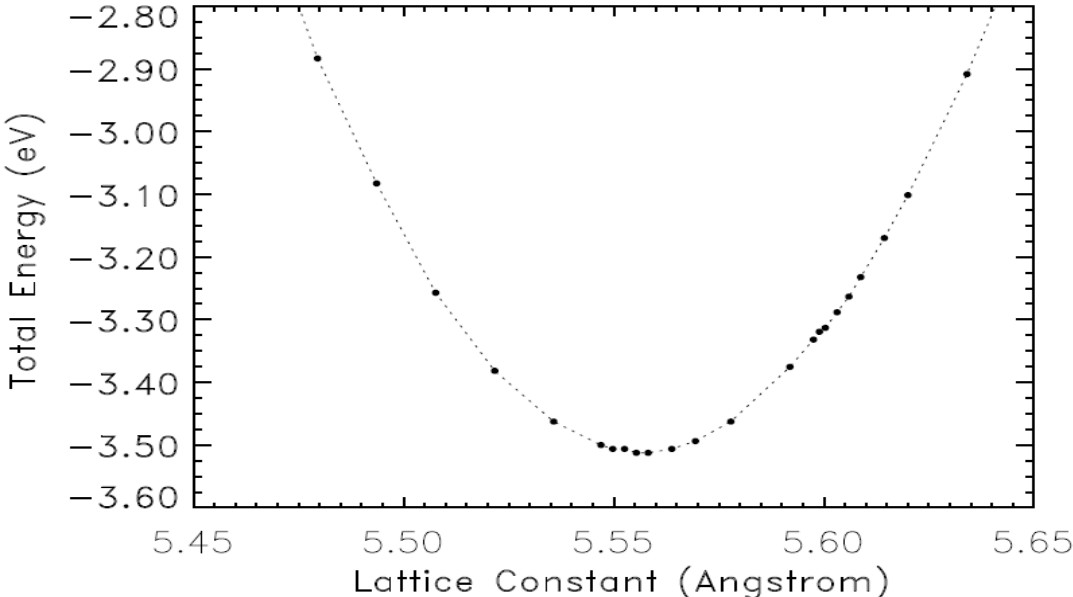

**Figure 3.** Variation of the total energy versus the lattice constant for zb-MgS.

## 4. Discussion

Our calculations produced the bandgap of 4.43 eV, which agrees with the experimental band gaps of 4.4 eV and 4.45 ± 0.2 eV. We obtained the exact band gap due to our adherence to the second DFT theorem. Namely, our generalized minimization of the energy verifiably leads the ground state of the material, and avoids basic sets that are over-complete for the description of the ground state and lead to unphysical lowering of unoccupied states. This process is required for a correct implementation of the DFT result. In several previous publications [25,30–34], we have thoroughly explained the fact that a self-consistent calculation with a single basis set, irrespective of the judiciousness of the selection of that basis set, produces a stationary solution among an infinite number of such solutions. The chances for that solution (a) to describe the ground state of the material and (b) not to have resulted from the use of an over-complete basis set are practically zero. The use of basis sets that are over-complete for the description of the ground state leads to spurious lowering of some unoccupied energies from their values obtained with the optimal basis set. Those values, along with occupied energies, are the only ones that belong to the spectrum of the Hamiltonian [25]. Both the Rayleigh theorem [25,43] for eigenvalues and the second corollary of the first DFT theorem [25,31,33,43,44] explain the spurious nature of unoccupied energies lowered from their values obtained with the optimal basis set. The noted unphysically lowering of unoccupied energies, including some of the lowest laying ones, while the occupied energies do not change, is a plausible explanation of the quasi-universal underestimation of band gaps in the literature. This scenario stems from the general tendency, in single basis set calculations, to select relatively large basic sets in order to ensure completeness.

Our computed electron effective masses are clearly greater than results from previous calculations. This difference can be explained with the increase in the curvature of the lowest, unoccupied band in calculations that employ a single basic set. This basic set is often selected to be large in order to meet complete requirements. We presume that the single basis set turns out to be over-complete for the description of the ground state of the material, resulting in the noted unphysical lowering of the bottom of the conduction bands. The increased curvature, inherent to this extra lowering, means smaller electron effective masses as compared to the ones from our calculation with the optimal basis set.

As far as we can determine from the content of the corresponding articles, the previous, calculated band gaps in Table 1, obtained with ab-initio LDA or GGA potentials, did not result from a generalized minimization of the energy. This generalized minimization is required by the second DFT theorem. It verifiably leads to the ground state results. These results are different from those for an arbitrary, stationary state among an infinite number of such states. The differences between our ground state results and those for an arbitrary, stationary state vary with the single basic set utilized to obtain the latter. The reader is urged to consult Reference [25] where we have proven that we do not need to invoke self-interaction correction (SIC) or derivative discontinuity in order to obtain results in agreement with the experiment. In particular, in the discussion section [25], we have elaborated on the limitations of SIC and of the derivative discontinuity of the exchange-correlation functional.

As for the results from ad-hoc DFT potentials, their values depend on the parameters or other adjustments they entail. Hence, calculations with these potentials, irrespective of the popularity of some of them, have no predictive capabilities. A full DFT potential, one that is based on a strict adherence to the two theorems of DFT, has to be the functional derivative of an exchange-correlation energy.

## 5. Conclusions

We provided the DFT description of the true ground state electronic, structural, and transport properties of zb-MgS. Our generalized minimization of the energy led to this accurate description. Our calculated LDA band gap, at room temperature, unlike previous theoretical results obtained with ab-initio DFT potentials, is in excellent agreement with corresponding, experimental ones of 4.45 ± 0.2 eV and 4.4 eV. Additionally, our calculated, partial densities of states, in accord with the experiment, place the S-p at the top of the valence band and Mg-s and S-s at the bottom of the conduction band. As the case for several previous results from our group [25], future experiments will likely confirm our other results, including those for the bulk modulus and particularly for the effective masses.

**Author Contributions:** Conceptualization, U.B. and D.B. Methodology, D.B. Software, D.B. Validation, U.B., B.A.A., and D.B. Formal analysis, Y.M. Investigation, Y.M. and B.A.A. Writing—original draft preparation, U.B. Writing—review and editing, U.B., L.F., and D.B. Supervision, D.B. Funding acquisition, D.B. All authors have read and agreed to the published version of the manuscript.

**Funding:** This work was funded partly by the National Science Foundation [NSF, Award Nos. EPS-1003897, NSF (2010–2015)-RH SUBR, and HRD-1002541], the U.S. Department of Energy, National Nuclear Security Administration (NNSA, Award No. DE-NA0002630), LaSPACE, and LONI-SUBR.

**Conflicts of Interest:** The authors declare no conflict of interest.

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
