# Peer review of "First Principle Investigation of Electronic, Transport, and Bulk Properties of Zinc-Blende Magnesium Sulfide"

_electronics, doi:10.3390/electronics9111791_

Round 1

Reviewer 1 Report

In this work Uttam Bhandari and co-workers reported the computational study on electronic, structural, and transport properties of zinc-blende magnesium sulfide (zb-MgS) by using the first principles method. The authors applied LDA-BZW-EF method to calculate and achieved the bandgap of zb-MgS is 4.43 eV with a given experimental lattice constant of 5.620 Å, which agrees well with the experimental results, but calculated electron effective masses are larger than other reported results. Other parameters like total (DOS) and partial (pDOS) densities of states of zb-MgS, equilibrium lattice constant at low temperature, and the bulk modulus are also reported in the work. In general, this is a very important scientific contribution, particularly from the development of computational methods and newly calculated property values to verify by future experiments. Additionally, the paper is well structured, exhibiting a valuable set of conclusions properly supported by the work performed throughout the text. I therefore recommend this work to be considered for acceptance in electronics, but just after performing a quick review in the form of minor revisions. I include some brief comments hereby:

  1. The authors may need to specify the low temperature to get the equilibrium lattice constant, as well as indicate more detailed conditions of calculating the bulk modulus.
  2. Please complete figure 3 caption text. The author may consider combine figure 2 and 3 together rather than present them in a two-column layout here for consistency.

Reviewer 2 Report

Dear Editor, dear Authors,

This manuscript presents interesting results of calculations of zb-MgS properties by DFT method. Authors obtained band structure of MgS and determined electron masses and energy gap. The calculations are state-of-the art. The only problem I can see is that the investigated material, MgS, is rarely used. The propositions of production of II-VI optoelectronic devices were not fulfilled. The II-VI wide band gap materials lost in competition to nitrides. However, lately there is interest in new 2D sulfides and may be the MgS layers will be compatible with these devices. Since MgS samples and related publications are rare, the authors predicted some parameters (e.g. electron mass) before the experiment could be made. It would be interesting to check these predictions. I think that the provided data could be very useful, so I recommend the publication.
Some corrections:
1) The experimental estimation of MgS bandgap Eg = 4.8 eV is presented in paper [5] and should be listed in table 1. It was obtained for structures containing MgS what is an advantage comparing to paper [22] that was based on measurements of ternary compounds.
2) Inset in figure 2 is described neither in caption nor in the text. Moreover, it is unreadable. I think, that the inset is just unnecessary.
3) In first lines of chapter 3.2 (line 173, 174) there is statement "charge mobilities". These are obviously mobilities of electrons and holes that are "charge carriers". I think that properly should be "charge carrier mobilities".
4) The obtained lattice constant (chapter 3.3) should be compared with experimental values 0.562 A [5] and 0.559 A [1].
5) The reference numbering is flown. I have checked that:
- References listed as [30-34] in line 63 are in fact [31,32,35] in the reference list.
- References listed as [35,36] in line 93 are in fact [36,37] in the reference list.
- Reference listed as [37] in lines 168 and 171 is in fact [38] in the reference list.
- The same for 38 -> 39, 39 -> 40 etc.
6) Informations in "Funding" and "Acknowledgements" are redundant (they are identical). It seems the one of them should be deleted.
